# Resistance to Targeted Agents Used to Treat Paediatric ALK-Positive ALCL

**DOI:** 10.3390/cancers13236003

**Published:** 2021-11-29

**Authors:** Lucy Hare, G. A. Amos Burke, Suzanne D. Turner

**Affiliations:** 1Department of Pathology, University of Cambridge, Cambridge CB2 0QQ, UK; lh709@cam.ac.uk; 2Department of Paediatric Oncology and Haematology, Addenbrooke’s Hospital, Cambridge CB2 0QQ, UK; amos.burke@addenbrookes.nhs.uk; 3CEITEC, Masaryk University, 62500 Brno, Czech Republic

**Keywords:** nucleophosmin1-anaplastic lymphoma kinase, anaplastic large cell lymphoma, resistance, chemotherapy, paediatric cancer

## Abstract

**Simple Summary:**

In general, the non-Hodgkin lymphoma (NHL), anaplastic large cell lymphoma (ALCL) diagnosed in childhood has a good survival outcome when treated with multi-agent chemotherapy. However, side effects of treatment are common, and outcomes are poorer after relapse, which occurs in up to 30% of cases. New drugs are required that are more effective and have fewer side effects. Targeted therapies are potential solutions to these problems, however, the development of resistance may limit their impact. This review summarises the potential resistance mechanisms to these targeted therapies.

**Abstract:**

Non-Hodgkin lymphoma (NHL) is the third most common malignancy diagnosed in children. The vast majority of paediatric NHL are either Burkitt lymphoma (BL), diffuse large B-cell lymphoma (DLBCL), anaplastic large cell lymphoma (ALCL), or lymphoblastic lymphoma (LL). Multi-agent chemotherapy is used to treat all of these types of NHL, and survival is over 90% but the chemotherapy regimens are intensive, and outcomes are generally poor if relapse occurs. Therefore, targeted therapies are of interest as potential solutions to these problems. However, the major problem with all targeted agents is the development of resistance. Mechanisms of resistance are not well understood, but increased knowledge will facilitate optimal management strategies through improving our understanding of when to select each targeted agent, and when a combinatorial approach may be helpful. This review summarises currently available knowledge regarding resistance to targeted therapies used in paediatric anaplastic lymphoma kinase (ALK)-positive ALCL. Specifically, we outline where gaps in knowledge exist, and further investigation is required in order to find a solution to the clinical problem of drug resistance in ALCL.

## 1. Introduction

### 1.1. Epidemiology and Pathologenesis of Paediatric ALCL

Anaplastic large cell lymphoma (ALCL) is a peripheral T cell non-Hodgkin lymphoma (NHL) with an annual incidence of 1.2 per million children aged under 15 [1]. The World Health Organisation (WHO) sub-classifies ALCL into anaplastic lymphoma kinase (ALK)-positive nodal/systemic, ALK-negative nodal/systemic, primary cutaneous and breast implant-associated ALCL [2]. The majority of paediatric ALCL is ALK-positive, usually due to a t(2;5)(p23;q35) chromosomal translocation causing the expression of the oncogenic breakpoint product NPM1-ALK [1,3]. This translocation results in unregulated ligand-independent activation of ALK, a receptor tyrosine kinase [2,4], due to homodimerization and auto-phosphorylation of its kinase domain [5] (Figure 1). The oncogenic activity of NPM1-ALK has been proven in a variety of genetically modified murine models and is therefore considered the driving event in these malignancies [6,7,8,9]. In essence, NPM1-ALK is able to activate a number of pathways that confer the hallmarks of cancer on incipient tumour cells [10] including pathways that facilitate increased survival and reduced apoptosis, such as the mitogen-activated protein kinase (MAPK) [11], phosphoinositide 3-kinase (PI3K)-Akt [12], Janus kinase (JAK)-signal transducer and activator of transcription (STAT) [13] and phospholipase C gamma (PLCγ) pathways [14] (Figure 2).

However, while NPM1-ALK is necessary for the development of ALK-positive ALCL, its presence alone is not sufficient for lymphomagenesis given that approximately 1% of newborn babies carry the t(2;5)(p23;q35) translocation (as evidenced by the presence of the translocation in cord blood stem cells) but, the incidence of ALCL is orders of magnitude lower [19]. In keeping with multi-step carcinogenesis, ‘second hits’ might include antigen-induced T-cell receptor (TCR) signalling, or antigen-independent microenvironmental factors [2,7,11,20,21] (Figure 3).

The uncertainty surrounding the precise pathogenesis of ALCL is made more complex by the illusive ‘cell of origin’. Most ALCL tumour cells express CD4 (indicative of helper T cells, although CD3 expression is seen in <20% tumours), granzyme B, TIA-1 and perforin (indicative of cytotoxic T cells), CD30 (indicative of activated lymphocytes) and sometimes CD25 (indicative of activated lymphocytes or regulatory T cells when co-expressed with CD4), with the common denominator being a T cell [2,20,22,23]. However, gene expression profiles of ALCL not only share features with T helper 17 (Th17) cells [24], but also with early thymic progenitors or haemopoietic stem cells [25]. Whether this latter genetic signature is a consequence of the cell of origin or NPM1-ALK-induced activity is debatable. In support of the former, NPM1-ALK enables thymocytes to skip beta-selection in mice, and it has been observed that two-thirds of human ALCL tumours have TCR rearrangements that would not normally be permissive of successful thymic development suggesting that NPM1-ALK is active in the aberrant thymocytes of children carrying this translocation [7,20]. However, NPM1-ALK is also able to transform primary human peripheral T cells to mimic ALCL and activate a gene signature associated with stemness [26,27,28,29]. Nevertheless, additional events, beyond the expression of NPM1-ALK, are required for the development of ALK-positive ALCL. In summary, ALCL is a rare paediatric NHL whose oncogenic driver is NPM1-ALK. Its exact cell of origin and pathogenesis remains the subject of continued research.

### 1.2. Clinical Presentation and Management of Paediatric ALCL

Children with ALCL usually present with high-grade advanced-stage nodal and extranodal disease [30]. Up to 61% have bone marrow involvement when minimal disseminated disease (MDD) is assessed using qualitative reverse-transcription polymerase chain reaction [31], and 5% have central nervous system (CNS) involvement [32]. Symptoms vary with disease location but typically include lymphadenopathy and B symptoms [30].

Paediatric ALCL is treated in many countries with the internationally recognised ALCL99 multi-agent chemotherapy regimen which has a 10-year overall survival (OS) of over 90% [20,33] but often leads to acute toxicities and late effects [34]. Additionally, progression-free survival (PFS) is only 70% and outcomes are poorer for relapsed cases [20,33]. The European Inter-Group for Childhood Non-Hodgkin Lymphoma (EICNHL) ALCL-RELAPSE trial suggested vinblastine monotherapy was effective for low-risk first relapse, and multi-agent chemotherapy followed by allogeneic stem cell transplant (SCT) was best for high-risk first relapse [2,35]. However, there is no absolute management consensus, particularly for subsequent relapses [20,33].

Current challenges in the treatment of paediatric ALCL are to find less toxic treatments, predict relapse, and treat it more effectively. Vinblastine monotherapy is minimally toxic and effective for a subgroup of patients with relapsed disease; it is the subject of an upcoming trial in newly diagnosed non-high-risk ALCL patients.

Vinblastine is a vinca alkaloid that binds to the mitotic spindle and stops cell division at metaphase. It is usually given at a weekly dose of 6 mg/m^2^ intravenously and is generally well tolerated, with the advantage that monotherapy can be delivered on an outpatient basis [36]. Vinblastine has shown efficacy in the treatment of relapsed/refractory ALCL whereby a study conducted by the French Society of Paediatric Oncology reported that vinblastine monotherapy (median duration 14 months) led to complete remission (CR) in 83% of relapsed patients, which was sustained for 7 years in 36% of cases [37]. Additionally, the EICNHL-ALCL-RELAPSE trial showed that 2 years of vinblastine monotherapy achieved a 3-year event-free survival (EFS) of 85% and OS of 90% for low-risk relapses [35]. This suggests that vinblastine-induced remissions are better sustained with longer treatment. However, in the first-line setting the efficacy of vinblastine is not as clear cut; EICNHL together with the Children’s Oncology Group (COG) investigated the addition of 1 year of vinblastine to standard first-line multi-agent chemotherapy for paediatric ALCL. Unfortunately, this approach did not reduce the relapse rate, and instead increased acute toxicity [38,39]. These data suggest that vinblastine is best utilised as a single agent if the goal is to reduce treatment-related toxicity. Therefore, EICNHL will investigate vinblastine monotherapy as a first-line treatment for low-risk patients, but with a longer treatment duration of 2 years in an upcoming trial [20]. Given the efficacy of the ALCL99 regimen, this is unlikely to further improve survival rates but is expected to reduce toxicity [33].

Resistance to vinblastine is a distinct possibility as has been reported for other malignancies. The major cause of vinblastine resistance in these cases is due to upregulation of expression of members of the ATP-binding cassette (ABC) transporter superfamily, such as ABCC1 and MDR-1 (P-glycoprotein). These transporters facilitate multi-drug resistance through their ability to efflux drugs, resulting in a reduced intracellular drug concentration. They have been shown to mediate vinblastine resistance specifically in renal cell carcinoma [40,41] and T cell lymphoblastic leukaemia [42] cell lines. Increased expression of these efflux pumps generally occurs through gene amplification or via downregulation of negative regulators of their expression, such as a decrease in MiR-210-3p which is a negative regulator of ABCC1 [40,43]. Many drugs have been investigated as potential therapies to overcome vinblastine resistance caused by ABC transporters. These include chloroquine, chlorpromazine, verapamil, cyclosporine A, quinine, valspodar, biricodar and zosuquidar [41,44,45,46,47]. However, these drugs did not perform well in clinical trials due to poor efficacy and/or toxicity problems and none are currently approved for the reversal of drug resistance caused by ABC transporters [46,47]. Indeed, through this mechanism, resistance to multiple standard toxic chemotherapy agents is induced and with the advent and increasing use of novel targeted agents, the search for inhibitors of multi-drug resistance in this form may become of decreasing necessity.

A second cause of vinblastine resistance is increased expression or activity of c-Jun, a member of the transcription factor, activator protein-1 (AP-1) family, which disrupts the ability of vinblastine to initiate apoptosis [48]. As AP-1 transcription factors are actively transcribed in ALCL, this mechanism of resistance is a distinct possibility [16]. A third cause of resistance is an alteration in tubulin content and polymerisation status, which interferes with the site of action of vinblastine [49]. However, despite these possible mechanisms, resistance to vinblastine monotherapy is not particularly common in ALK-positive ALCL, as it remains effective for subsequent relapses that occur after therapy cessation in the majority of patients [37]. Additionally, as previously mentioned, the increased use of novel targeted agents will likely negate any pressure to develop counter-active measures towards resistance to vinblastine.

Emerging agents targeting the pathways dysregulated by NPM1-ALK hold great promise for both newly diagnosed and relapsed patients. However, both primary (intrinsic) and acquired resistance to these drugs are a major problem and, at present clear solutions or guidelines to overcome this issue do not exist. A prerequisite for establishing solutions is a detailed understanding of the potential mechanisms of resistance and therefore this review will focus on the latter rather than providing solutions.

## 2. Targeted Agents for the Treatment of ALCL

Studying targeted agents for the treatment of paediatric ALK-positive ALCL is challenging due to the low disease incidence. However, coordinated international efforts to study the treatment of paediatric ALCL, in addition to knowledge gained from investigating adult ALCL and other ALK-positive diseases, has led to the suggestion that ALK tyrosine kinase inhibitors (ALK TKIs), armed antibodies to CD30 (such as brentuximab vedotin (BV)) and immune checkpoint inhibitors (such as nivolumab) may be useful drugs to overcome the current challenges in the treatment of paediatric ALCL.

### 2.1. ALK Tyrosine Kinase Inhibitors

ALK TKIs inhibit the kinase activity of aberrantly expressed ALK largely through binding to the ATP pocket. They are given orally and are generally well tolerated as monotherapy with few, largely manageable side effects [20,50]. As ALK is usually only expressed in neonatal neurons, off-target side effects were expected to be minimal [51], particularly as ALK knockout mice are viable without any obvious health defects [52,53]. The use of ALK TKIs to treat ALCL has benefitted from prior investigations of their use in ALK-positive non-small cell lung cancer (NSCLC) [1,54]. The first ALK TKI approved for NSCLC by the Food and Drug Administration (FDA) was crizotinib, but second (ceritinib, alectinib and brigatinib) and third (lorlatinib) generation inhibitors have subsequently also been authorised [1]. Approval for use in ALCL has been difficult, particularly in children due to disease rarity [20]. However, crizotinib recently gained FDA approval for relapsed/refractory ALK-positive ALCL in patients aged over one year [55]. This was based on evidence that crizotinib was well tolerated and led to a complete response in 67–83% of patients with relapsed ALCL [50,56,57]. Additionally, alectinib has recently gained approval in Japan for relapsed ALCL, due to a trial showing efficacy (PFS 58.3%, EFS and OS both 70%) [58].

How ALK TKIs should be combined with or replace existing frontline therapy remains to be resolved given the already high survival rate of multi-agent chemotherapy protocols such as ALCL99. A recently completed trial (NCT01606878) employed crizotinib in combination with ALCL99 therapy and showed unacceptable cytopenias and gastrointestinal toxicity in the paediatric ALCL relapse setting [59]. Another trial (ITCC053) closed the arm in which crizotinib and vinblastine were given in combination for relapsed paediatric ALCL due to toxicities [60]. The full results of an additional trial (NCT01979536) administering crizotinib with the ALCL99 chemotherapy regimen in newly diagnosed paediatric ALCL, are awaited, although a temporary pause in the trial had to be imposed due to a number of thromboses [1,61]. Whether other ALK inhibitors will result in different toxicity profiles remains to be seen. An industry-led trial (Takeda) combining brigatinib with ALCL99 multi-agent chemotherapy is planned. Brigatinib has shown efficacy in adults with NSCLC resistant to crizotinib and unlike crizotinib, brigatinib has shown good brain penetrance and durable responses in patients with NSCLC [62], and it is hoped that its use in the treatment of ALCL will be similarly effective.

### 2.2. Brentuximab Vedotin

All ALK-positive paediatric ALCL express the transmembrane receptor CD30, which facilitates cancer cell survival through the activation of, for example, the nuclear factor-kappa B (NF-kB), Interferon regulatory factor 4 (IRF4) and MYC proto-oncogene (MYC) pathways [63,64]. However, monoclonal antibodies (MABs) targeting CD30 alone were disappointing; only 17% of adult ALCL patients had any response to the anti-CD30 MAB SGN-30 in a Phase II trial [65]. Therefore, the antibody-dug conjugate BV was developed. This consists of an antibody targeting CD30 bound to the anti-tubulin agent monomethyl auristatin E (MMAE). When the antibody binds to CD30, the drug-receptor complex is internalised, and after lysosomal processing, MMAE is released inside the cell, driving apoptosis [63].

BV was approved by the FDA for adults with relapsed ALCL in 2011 [1]. This was based on early results from a phase II study (NCT00866047) which has since demonstrated durable remissions at five years [66,67]. BV was next recommended for use alongside multi-agent chemotherapy in newly diagnosed adult ALCL due to the results of the phase III ECHELON-2 trial [68]. Subsequently, BV has been studied for use in paediatric ALCL. A phase I/II study (NCT01492088) including paediatric patients with relapsed/refractory ALCL showed that the pharmacokinetic profiles were similar to those in adults; 53% of patients achieved an overall response, and side effects were manageable [69]. Another phase II study (ANHL12P1) showed that the addition of BV to the ALCL99 chemotherapy regimen for newly diagnosed ALK-positive paediatric ALCL led to an EFS of 79.1%, an improvement from the assumed EFS of 70% for ALCL99 therapy alone, without any additional toxicity. Moreover, this treatment approach almost completely eliminated progression/relapse during first-line treatment. This is particularly important because children who progress during first-line treatment have the poorest prognosis [63,70].

### 2.3. Checkpoint Inhibitors

ALK-positive ALCL cells constitutively express the immune checkpoint protein programmed cell-death ligand 1 (PD-L1) on their surface. This interacts with the programmed cell death 1 (PD-1) receptor to dampen the T cell immune response against the tumour, although the precise mechanism via which this occurs is currently under investigation [71,72]. MABs directed against PD-L1, nivolumab and pembrolizumab, may be particularly effective in ALK-positive ALCL because NPM1-ALK drives high levels of PD-L1 expression via STAT3 [1]. Indeed, initial case reports describe durable responses and limited toxicities in teenagers with relapsed/refractory ALK-positive ALCL [73,74]. Patients are currently being recruited to a phase II trial (NCT03703050, NIVO-ALCL) to study the use of nivolumab in adult and paediatric patients with relapsed/refractory ALK-positive ALCL previously treated with chemotherapy and either an ALK TKI or BV [1].

## 3. Mechanisms of Resistance to ALCL Therapy

It is anticipated that resistance to these novel agents may be a major problem. Understanding how this resistance develops, and how it can be overcome, will be essential to the successful integration of these novel approaches into the mainstream treatment of paediatric ALK-positive ALCL.

### 3.1. Mechanisms of Resistance to ALK Tyrosine Kinase Inhibitors

Despite the promise of ALK TKIs, it appears at least in some contexts, that therapy will either have to be for long periods of time or as a bridge to another treatment, as swift relapses have been observed on the discontinuation of crizotinib monotherapy [1,75]. In addition, resistance is expected as has been experienced for patients with ALK-positive NSCLC [17]. Indeed, reports of crizotinib resistance developing within months of treatment initiation in patients with ALCL have been published [56,57].

ALK TKI resistance can either be ALK-dependent or -independent with the former largely occurring due to mutations in ALK (Figure 1 and Figure 4). These have been reported as either mutations in residues at the TKI binding site, or those that result in conformational changes that increase aberrant ALK activity [76,77]. Extensive studies of ALK-dependent resistance mechanisms in NSCLC [17,78] have identified numerous mutations which in some cases are specific to a certain ALK TKI and in others are ubiquitous amongst the ALK TKIs (Table 1). In the treatment of NSCLC, clinicians usually start by treating with crizotinib, and then swap to second- or third-generation inhibitors depending on the ALK mutation that has developed. Ultimately this increases the risk of compound mutations developing, which confer resistance to second and even third-generation inhibitors [17,79]. However, some of these compound mutations can actually re-sensitise patients to crizotinib, even if it has been administered to the patient previously [17]. Cycling of TKIs may therefore be an approach to preventing or responding to the development of resistance.

Another form of ALK-dependent resistance is caused by the amplification of the ALK gene (Figure 4). This results in resistance due to a target excess [17,78] as has been reported for NSCLC cell lines [80,81] and patients [78,82,83], as well as ALCL cell lines [84,85] but not yet for ALCL patients. Interestingly, ALK upregulation in response to ALK TKIs results in overwhelming ALK signalling such that if the ALK TKI treatment is stopped, the excess in ALK signalling can counterintuitively drive apoptosis. Therefore, a non-continuous dosing regimen may be beneficial [85,86,87].

**Table 1 cancers-13-06003-t001:** Summary of reported ALK mutants conferring resistance to ALK TKIs. Sites of the identified mutations are reported according to their position in the full-length ALK protein. Resistance mutations refer to those reported in the context of conferring resistance to ALK TKIs, some of which have been proven to confer sensitivity to ALK TKIs and others with conflicting evidence as to their ALK TKI response. Ins = insertion, del = deletion.

ALK TKI	ALK Mutation		
	Resistance	Sensitivity	Conflicting
Crizotinib	C1156T [88] C1156Y [17,79,89,90,91,92,93,94]	L1198F [79,89,95,96] C1156Y/L1198F [95]	
	D1203N [79,96,97,98]	G1202R/L1198F [91]	
	E1210K [79,90,99]	I1171N/L1265F [91]	
	F1174C [76,79,89,91,99] F1174I [91] F1174L [100,101] F1174V [91] F1245V [92]		
	G1128A [102]		
	G1202R [78,79,89,90,94] G1202del [79,89]		
	G1269A [79,82,89,90,94,96,99] G1269S [103]		
	I1171N [79,89,90,103,104,105] I1171S [79,89,90,103] I1171T [76,79,89,90,91,92,98] I1171X [106]		
	I1268L [91]		
	L1152P [103,107]		
	L1152R [81,94]		
	L1196M [17,78,79,80,82,89,90,91,92,93,94,96,98,99] L1196Q [91,96,104]		
	L1198P [97]		
	R1192P [108]		
	S1206C [90,103] S1206Y [78,79,90,94]		
	T1151K [109,110] T1151M [108]		
	V1180L [96]		
	Q1188_L1190del [111]		
	1151Tins [78,90]		
	D1203N/E1210K [79,89]		
	D1203N/F1174C [79,89]		
	F1174L/G1269A [98]		
Ceritinib	F1174L [79,90,103,107,108]	E1210K [79,89]	C1156Y [79,89,90,103,107,112]
	F1174S [90]	F1245C [113]	D1203N [79,89,90,96]
	F1174V [103]	I1171T [79,89,107]	F1174C [79,89,107,112]
	G1123S [114]	I1268L [91]	G1202R [79,89,90,92,98,103,112]
	G1128A [98]	L1196Q [91]	G1269A [79,89,96,98,107,108]
	G1202del [79,89,103]	S1206Y [107,115]	I1171N [79,89,90,107]
	L1122V [96]	V1185L [91]	I1171S [79,89,90]
	L1152P [103]	G1269A/I1171S [116]	L1196M [79,89,90,91,96,107,115]
	L1152R [103,107]	G1269A/I1171N [91]	
	L1198F [79,89,96]		
	R1192P [108]		
	R1275Q [17]		
	T1151K [109]		
	T1151M [108]		
	T1151Sins [117]		
	Q1188_L1190del [111]		
	1151Tins [103,107]		
	C1156Y/I1171N [79]		
	C1156Y/G1202del/V1180L [79]		
	D1203N/E1210K [79,89]		
	D1203N/F1174C [79,89]		
	E1210K/I1171T [98]		
	G1202R/F1174L [79]		
	G1202R/F1174V [92]		
	G1202R/L1196M [91]		
Alectinib	F1174I [91]	C1156Y [79,89,118]	F1174C [76,79,89]
	F1174L [79,90,91,103,107,108]	D1203N [79,89,96]	I1171T [76,79,89,90,91,119]
	F1174S [90]	E1210K [79,89]	L1196M [79,89,90,91,96,118]
	F1174V [91,103]	F1174L [118]	1151Tins [118]
	G1202R [79,89,90,92,99,118]	G1269A [79,89,96,118]	
	G1202S [99]	I1268L [91]	
	G1202del [79,89]	L1152R [118]	
	G1210K [120]	L1198F [79,89]	
	G1269A [108]	L1256F [91]	
	I1171N [79,89,90,91,92]	S1206Y [118]	
	I1171S [79,89,90,91,108]	T1151K [110]	
	I1171 X [106]	V1185L [91]	
	L1122V [96]	I1171N/L1256F [91]	
	L1196Q [91]		
	L1198F [96]		
	R1192P [108]		
	T1151M [108]		
	V1180L [119]		
	W1295C [98]		
	D1203N/E1210K [79,89]		
	D1203N/F1174C [79,89]		
	F1174L/G1269A [98]		
	G1202R/L1196M [98]		
	L1196M/V1185L [91]		
Brigatinib	G1202L [121]	C1156Y [79,89,107]	D1203N [79,89,90,96,107]
	G1202del [79,89]	F1174C [79,89,107]	E1210K [79,89,90]
	L1122V [84,96]	F1174L [107]	G1202R [62,79,89,90,107]
	S1206C [90]	G1269A [96,107]	I1171N [79,89,104,107]
	D1203N/F1174C [89]	I1171S [79,89,106]	L1198F [79,89,96,107]
	D1203N/E1210K [89]	I1171T [79,89]	S1206Y [17,90,107]
	E1210K/S1206C [89,99,103]	L1152P [107]	
	F1174V/L1198F [84]	L1152R [107]	
	F1174L/L1198V [99]	L1196M [79,80,89,96,107]	
	G1202R/L1196M [122]	L1196Q [104]	
		V1180L [96,107]	
		1151Tins [107]	
		G1269A/I1171S [116]	
		G1269A/I1171N [91]	
		I1171N/L1196M [91]	
		I1171N/L1198F [91]	
		I1171N/L1256F [91]	
Lorlatinib	C1156F [123]	C1156Y [79,89]	E1201K [79,89,123]
	G1128S [123]	D1203N [79,89]	G1269A [79,89,96,124]
	L1256F [91]	F1174C [79,89]	I1171N [79,89,123]
	C1156F/D1203N [123]	F1174I [123]	I1171T [79,89,123]
	C1156F/L1198F [124,125]	F1174L [126]	
	C1156Y/D1203N [125]	F1245C [126]	
	C1156Y/F1174C [125]	G1202del [79,89]	
	C1156Y/F1174I [125]	G1202K [120]	
	C1156Y/F1174V [125]	G1202L [121]	
	C1156Y/G1269A [125,127]	G1202R [79,89,128]	
	C1156Y/I1171T [125]	I1171S [79,89]	
	C1156Y/L1196M [125]	L1196M [79,89,96]	
	C1156Y/L1198F [125]	R1275Q [126]	
	C1156Y/S1256F [125]	V1180L [96]	
	D1203N/F1174C [79,89]		
	D1203N/L1196M [127]		
	F1174C/G1202R [91]		
	F1174C/G1269A [125]		
	F1174C/L1196M [125]		
	F1174L/G1202R [91,127]		
	G1202R/G1269A [91,98,124]		
	G1202R/I1171N [91]		
	G1202R/L1196M [92,122,125]		
	G1202R/L1198F [91,125]		
	G1269A/I1171S [116]		
	G1269A/I1171N [125]		
	G1269A/I1171T [125]		
	G1269A/L1196M [91,125]		
	G1269A/N1178H [124]		
	I1171N/C1156Y [91]		
	I1171N/L1198F [91]		
	I1171N/L1256F [91]		
	L1196M/F1174C [125]		
	L1196M/F1174L [125]		
	L1196M/F1174V [125]		
	L1196M/I1171S [125]		
	L1196M/I1179V [125]		
	L1196M/L1198F [125]		
	L1196M/L1198H [125]		
	L1196M/L1256F [125]		

ALK-independent mechanisms of resistance occur when the need for NPM1-ALK is bypassed through the activation of its downstream targets via alternative signalling cascades, so-called bypass tracks (Figure 5 and Table 2) [17,78]. For example, increased signalling through the insulin-like growth factor receptor (IGF-1R) activates the JAK/STAT, MAPK and PI3K/Akt pathways normally stimulated by EML4-ALK or NPM1-ALK, causing crizotinib resistance in NSCLC and ALCL respectively [129,130].

Alternatively, ALK-independent resistance can be induced due to activation of the pathways downstream of aberrant ALK activity via mutation of genes encoding proteins involved in these pathways (Figure 4). For example, mutations in the MAPK pathway components KRAS [82,99,131,144,145], NRAS [79,98], BRAF [79,98,132,140] and MEK [132,146], as well as reduced DUSP6 (a MAPK phosphatase) [144] and mutations in the RAS negative regulators neurofibromin 1 and 2 (NF1/2) [99,127], cause ALK TKI resistance in NSCLC. Furthermore, lorlatinib and ceritinib resistance has been associated with upregulation of the MAPK pathway in ALCL xenografts [124] and neuroblastoma cell lines [147], respectively. Additionally, PI3KCA mutations cause ALK TKI resistance in NSCLC [79,98,132,146], and lorlatinib resistance of ALCL xenografts has been associated with PI3K/Akt pathway upregulation [124]. Finally, NOTCH1 mutations in NSCLC [99] and ALCL [148], and PIM1 overexpression in neuroblastoma [149], also cause ALK TKI resistance through their effects on the JAK-STAT pathway. In a similar manner, we have shown that activation of signalling via the IL10R bypasses NPM-ALK to activate STAT3 in ALCL mediating resistance to crizotinib, alectinib, brigatinib and lorlatinib [142].

Besides ALK-dependent and independent mechanisms of resistance, there are some additional mechanisms through which ALK TKIs cease to be effective. Some are well studied, such as the CNS relapses that occur if crizotinib and ceritinib are effluxed from the CNS by P-glycoprotein in the blood-brain barrier. These effects can be overcome by using alectinib, brigatinib or lorlatinib as they are not substrates for P-glycoprotein [107]. Other resistance mechanisms are very specific to individual tumour types. For example, NSCLC acquires resistance through epithelial–mesenchymal transition, whereby cells lose their polarity and become more fibroblastic and invasive [79,127,150,151]. NSCLC also acquire resistance through transforming to a small cell lung cancer (SCLC) phenotype, although given that these cells retain their ALK rearrangements, further investigation is required to determine why these transformations mediate resistance [152,153,154,155]. Additionally, neuroblastoma cells driven by MYCN and ALK amplifications can acquire resistance via a multi-step process in which they downregulate MYCN, and instead upregulate and become dependent on BORIS, a DNA binding protein that increases the proliferation and survival of cancer cells, and here causes increased expression of transcription factors that promote the transformation to an ALK TKI-resistant phenotype [156,157]. Further resistance mechanisms are less well studied, such as the generation or loss of different ALK fusions. Indeed, in the case of NSCLC with 3 co-existing rare ALK fusions at diagnosis (COX7A2L-ALK, LINC01210-ALK and ATP13A4-ALK), the generation of an additional SLCO2A1-ALK fusion mediated crizotinib resistance, and the subsequent loss of the ATP13A4-ALK and SLCO2A1-ALK fusions mediated ceritinib resistance [158]. Additionally, increased autophagy in which cells break down obsolete constituents of their own cytoplasm as a way of generating additional energy for tumour growth and proliferation leads to crizotinib resistance in ALCL [159] and NSCLC [160] by allowing the cell to overcome the metabolic stress of the ALK TKI. Indeed, treatment of ALCL cells (both in vitro and in vivo) with crizotinib and chloroquine, which inhibits autophagy, resulted in a greater inhibitory effect than treatment with crizotinib alone, suggesting that there are potential ways of overcoming resistance caused by this mechanism [159].

Another potential cause of resistance to ALK TKIs is p53 disruption. It is already known that *TP53* mutation can cause resistance to traditional genotoxic chemotherapeutic agents by preventing apoptosis despite chemotherapy-induced DNA damage [161,162]. However, the resulting genetic instability caused by p53 disruption enables mutations to accumulate over time, and it is these later changes that may drive resistance to targeted agents. This has been shown to be the case in chronic lymphocytic leukaemia [161,162]. Additionally, in neuroblastoma, it has been shown that combination treatment with an ALK TKI and a p53 activator may prevent the resistance seen with ALK TKI monotherapy because the combination stimulates apoptosis rather than reversible growth arrest seen in cells treated with monotherapy [163]. However, p53-mediated resistance to targeted agents has not yet been demonstrated in ALCL. This should be investigated further because p53 is inactivated in some cases of ALCL, occasionally due to *TP53* gene mutations [164] but more usually via NPM1-ALK stimulated induction of JNK and MDM2 activity [165]. Additionally, it has been shown that the p53 activator nutlin-3a can induce apoptosis of ALCL and thereby enhance the efficacy of chemotherapy [166].

In summary, due to a large number of possible ALK TKI resistance mechanisms, it is essential that tumours are assessed at the time of resistance to identify the underlying cause, to inform on the next best therapeutic approach towards a cure. For example, the identification of an ALK mutation could dictate which ALK TKI to use next, and the emergence of ALK-independent bypass mechanisms may identify other druggable targets to overcome resistance. Indeed IGF-1R [129], HER [99], HGF [139], SRC [146], MEK [146] and mTOR [167] inhibitors have been shown to overcome ALK TKI resistance caused by activation of these bypass pathways. Therefore, these could be considered as future treatments in these cases. It is also imperative that further potential resistance mechanisms continue to be identified in the laboratory as for many patients, the mechanisms cannot be explained by known pathways. In evidence, a large study of NSCLC patients resistant to ALK TKIs found that only 33–44% of the resistance phenotype could be explained by currently known resistance mechanisms [98]. This is even more important for ALCL, as well as neuroblastoma, because the majority of known ALK TKI resistance mechanisms have been studied in NSCLC and may differ substantially between diseases due to differing underlying biology and activities specific to the type of aberrant ALK expression observed, including overexpression of full-length ALK, compared to a fusion protein.

Additionally, the schedule of ALK TKI delivery and the combination of drugs with which it is administered may also impact resistance mechanisms. For example, resistance may be delayed or prevented when combination therapies are given upfront rather than as sequential monotherapies; when therapy is given in a metronomic manner; or when treatments are cycled before resistance has developed rather than changing treatment once resistance has already developed. In particular, alternating ALK TKIs based on the additional proteins that they target other than ALK may be useful in reducing the selective pressure leading to specific ALK mutations and bypass resistance tracks. For example, crizotinib (also an MET and ROS1 inhibitor [90]) could be cycled with alectinib (also an RET, LTK and GAK inhibitor [90]). Furthermore, it has been demonstrated in NSCLC that MET-driven ALK TKI bypass resistance is present at lower levels in patients treated with a less selective ALK TKI [168]. This requires investigation specific to ALCL but suggests that the development of drugs with potent ALK inhibition, but not necessarily more ALK specificity, is required.

### 3.2. Mechanisms of Resistance to Brentuximab Vedotin

Resistance to BV develops relatively frequently, with around half of patients with relapsed/refractory ALCL treated with BV either progressing on therapy or requiring additional treatments [169]. The most obvious mechanism of resistance is a reduction in CD30 protein expression, the target of BV activity. This has been observed in at least one case of adult ALK-negative ALCL [170] and in epithelioid inflammatory myofibroblastic sarcoma patient-derived xenografts (PDXs) [171]. It is unclear whether these reductions in CD30 were due to downregulation of the CD30 target, increased turnover of CD30, CD30 internalisation, or the selective outgrowth of sub-clones with lower levels of CD30 expression [171]. However, CD30 target reduction does not account for all BV resistance because CD30 expression is maintained in some BV-resistant Hodgkin lymphoma and ALCL [172]. A second potential resistance mechanism is upregulation of ABC transporters, which has been demonstrated in BV-resistant Hodgkin lymphoma cell lines and patient samples [172,173,174], and epithelioid inflammatory myofibroblastic sarcoma PDXs [171]. However, this also cannot account for all BV resistance because Hodgkin lymphoma cells treated with BV and an MDR-1 inhibitor eventually stop responding to this combination [174].

In summary, further work is required to fully elucidate how to use BV in paediatric ALK-positive ALCL. In order to explore whether BV might allow traditional chemotherapy to be reduced, a better understanding of BV resistance mechanisms is required. This will help to establish whether it can be used as a monotherapy, with factors put in place to prevent the acquisition of resistance, or whether it should only ever be used alongside other agents.

### 3.3. Mechanisms of Resistance to Immune Checkpoint Inhibitors

Resistance to anti-PD-L1 immune checkpoint inhibitors will eventually occur in the majority of patients, although there are differences in rates between tumour types. This can be a consequence of PD-L1 upregulation [175,176,177], which can occur due to increased JAK-STAT signalling, caused by: loss of the tumour suppressor FBP1 [176,178], mutations in JAK1/2 [179], or stimulation of tumour cells’ interferon-gamma receptor 2 by interferon-gamma released from activated CD8^+^ T cells in the tumour microenvironment [180,181]. PD-L1 upregulation can also occur when loss of the tumour suppressor PTEN results in increased PI3K/Akt signalling [176,177].

Alternatively, resistance to anti-PD-L1 immune checkpoint inhibitors can occur as a result of a skew in the distribution of immune cells in the tumour microenvironment towards an immunosuppressive one, compensating for the effects of loss of PD-L1 signalling. This skew can be caused by a low overall T cell abundance [181], an increased ratio of regulatory T cells to effector T cells [182], dysfunctional CD8^+^ T lymphocytes [183,184], increased myeloid-derived suppressor cells (MDSCs) [185] and/or increased PD-L1 positive M2 macrophages [185,186]. These changes are mediated in a variety of ways (Table 3), which might be therapeutically targetable. For example, a clinical trial (NCT03048500) is investigating whether metformin used alongside nivolumab in NSCLC can overcome the nivolumab resistance caused by the detrimental effects of hypoxia on CD8^+^ lymphocytes [184,187].

Another cause of resistance to anti-PD-L1 immune checkpoint inhibitors is a reduction in neoantigen expression on the tumour cell surface. A lack of neoantigens, either due to a low tumour mutational burden, or altered antigen processing and presentation, results in a reduced immune response to the tumour, which compensates for the immunosuppressive effects of PD-L1 signalling that have been lost [180,185,191].

It is important to note that although there is some understanding of mechanisms of resistance to immune checkpoint inhibitors, these may not all be relevant to ALCL. For example, there is an endogenous immune response to ALCL via the production of anti-ALK autoantibodies amongst other mechanisms, therefore, an immunosuppressive microenvironment and a lack of neoantigens may not be as relevant in ALCL as it is in other diseases [1]. Further research is required to determine whether these potential resistance mechanisms apply in ALCL, to find other resistance mechanisms, and to determine how to overcome these.

## 4. Conclusions

In summary, there is a trade-off between reducing side effects and the potential for resistance to develop when choosing targeted therapies for ALK-positive paediatric ALCL. Knowledge of potential resistance mechanisms is starting to develop but large gaps remain. Further investigation into how resistance develops is an essential prerequisite for finding a solution to this resistance problem so that targeted agents can be successfully integrated into the mainstream treatment of ALK-positive paediatric ALCL. This will include the investigation into potential differences between primary (intrinsic) resistance, where there is a lack of response to treatments from the start of therapy, and acquired resistance, where the selective pressure of exposure to a treatment results in the outgrowth of resistant sub-clones and eventual resistance after an initial response [17]. Overall, the ability to evaluate patient tumour samples for known resistance mechanisms prior to treatment will enable the prediction of treatment response, and after the development of resistance will enable identification of the causes. This will inform on the best therapeutic approach towards a cure on an individual patient basis.

## Figures and Tables

**Figure 1 cancers-13-06003-f001:**
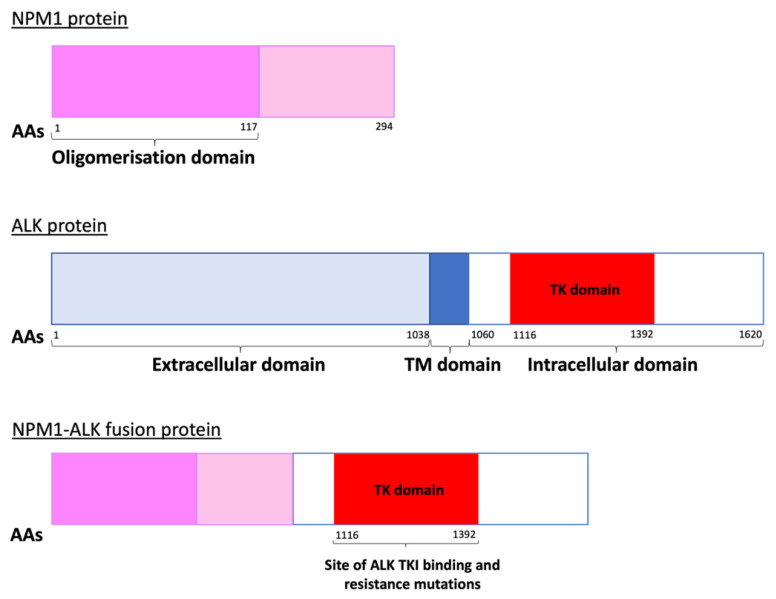
The NPM1-ALK fusion protein produced due to a t(2;5)(p23;q35) chromosomal translocation. The kinase domain, depicted in red, is the site within the ALK portion of the fusion protein where ALK tyrosine kinase inhibitors (ALK TKIs) bind. Mutations here can lead to ALK TKI resistance [15,16]. AAs = amino acids, TM = transmembrane, TK = tyrosine kinase.

**Figure 2 cancers-13-06003-f002:**
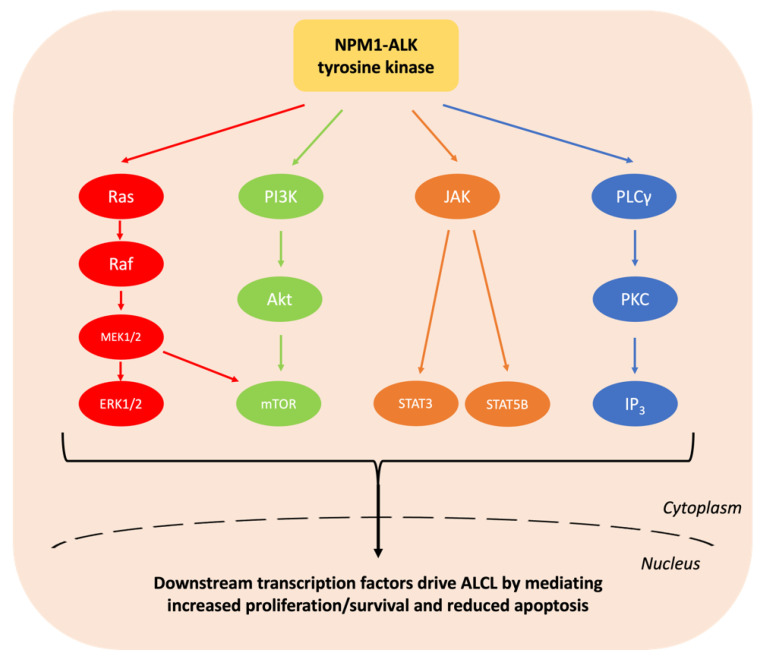
Summary of the signalling pathways activated by NPM1-ALK. The hyperactive intracellular tyrosine kinase NPM1-ALK activates a plethora of signalling pathways including MAPK, PI3K-Akt, JAK-STAT and PLCγ. These collectively drive ALCL through conferring the hallmarks of cancer including increasing cell survival and reducing apoptosis [11,12,13,14,17,18].

**Figure 3 cancers-13-06003-f003:**
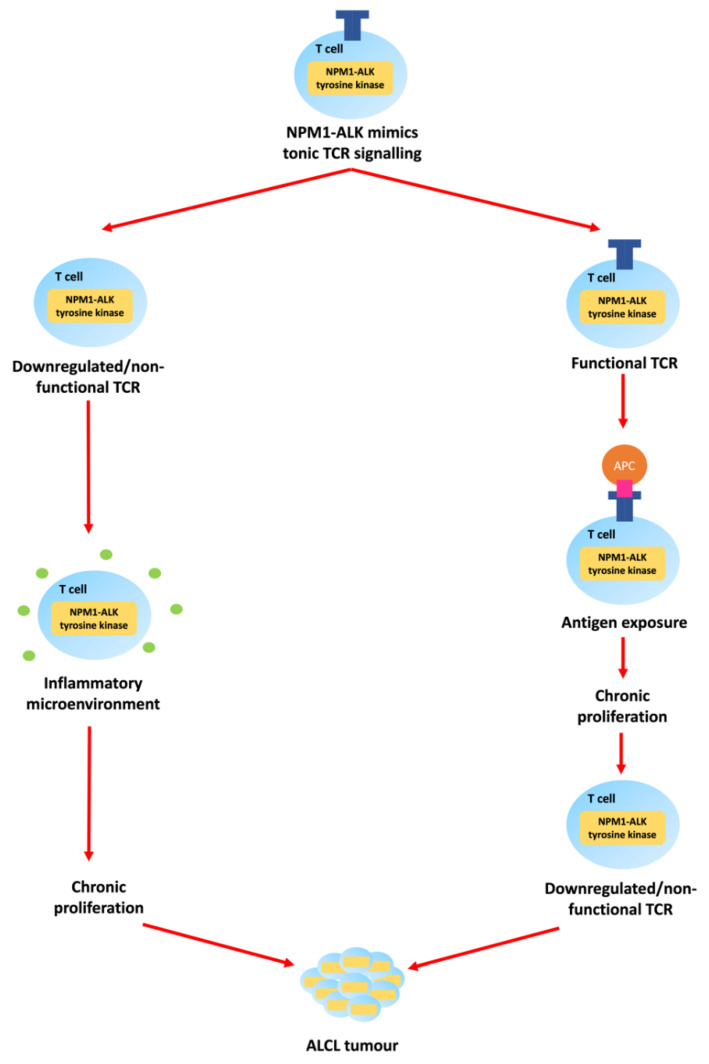
The pathogenesis and aetiology of ALCL expressing NPM1-ALK may be dependent on TCR signalling and/or microenvironmental factors. NPM1-ALK mimics low-intensity tonic TCR signalling required for T cell development. The TCRs are ultimately downregulated as they are either surplus to requirement or are prohibitive towards tumour development. If the TCR is downregulated or non-functional soon after emerging into the periphery, antigen-independent inflammatory microenvironmental factors might provide the ‘second hit’ promoting ALCL development. If the TCR is functional after emerging into the periphery, an antigen-presenting cell (APC) might expose the T cell to a ‘second hit’ in the form of a major histocompatibility complex (MHC)-bound ligand that provides additional stimulation and promotes ALCL development. With this additional stimulation, the TCR might then be downregulated to facilitate cell survival by preventing over-stimulation [2,7,11,20,21].

**Figure 4 cancers-13-06003-f004:**
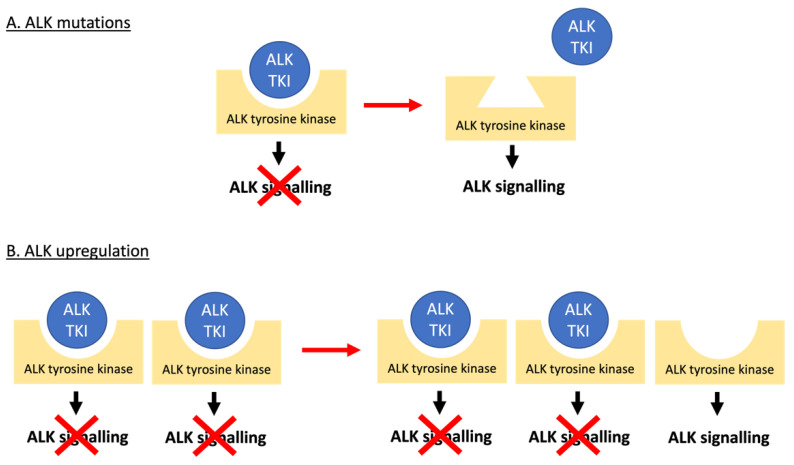
ALK-dependent mechanisms of resistance to ALK TKIs. (**A**) Mutations in the ALK tyrosine kinase domain prevent the ALK TKI from binding to the receptor and exerting its inhibitory effect on oncogenic ALK signalling. (**B**) Amplification of the *ALK* gene provides an excess of drug target outcompeting the inhibitor.

**Figure 5 cancers-13-06003-f005:**
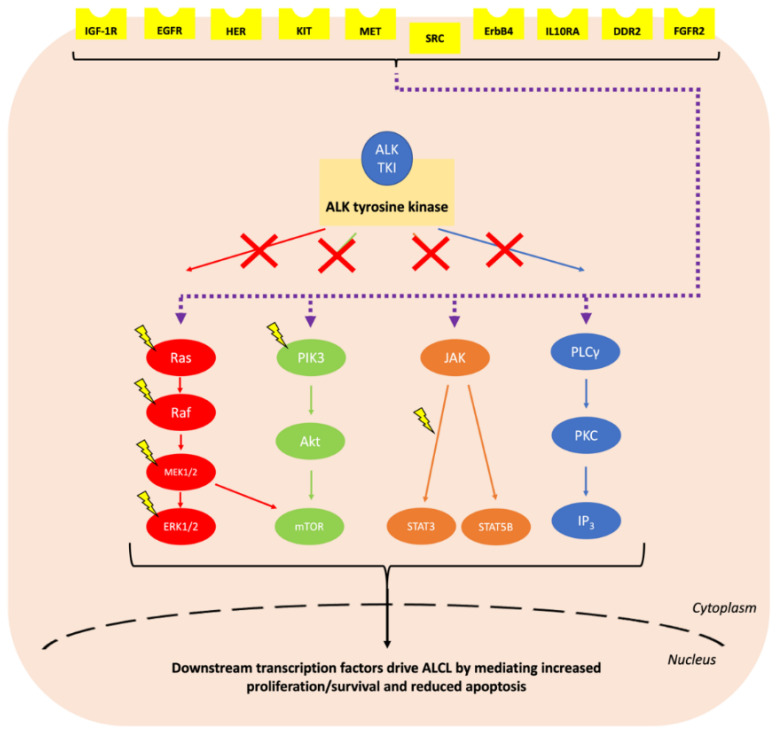
ALK-independent ‘bypass’ mechanisms of resistance to ALK TKIs. The effects of inhibited ALK activity are substituted by the upregulation of alternative signalling cascades that activate the same downstream targets as does aberrantly active ALK. The need for ALK is effectively bypassed. Mutations in the downstream targets of ALK, indicated with a lightning symbol, also bypass the need for ALK.

**Table 2 cancers-13-06003-t002:** Summary of reported bypass tracks conferring resistance to ALK TKIs.

Protein Alteration (Upregulation Unless Otherwise Specified)	ALK TKI	Disease
IGF-1R	Crizotinib [129,130]	NSCLC and ALCL
Epidermal growth factor receptor (EGFR)	Crizotinib [78,81,98,131,132] Ceritinib [98,133] Alectinib [98,132,134] Lorlatinib [124,132]	NSCLC
	Lorlatinib [124]	Neuroblastoma
Human epidermal growth factor receptor (HER), including via increased neuregulin 1 ligand	Ceritinib and alectinib [135,136]	NSCLC
KIT proto-oncogene receptor tyrosine kinase (KIT), including via increased stem cell factor (SCF) ligand	Crizotinib [78] Ceritinib [98]	NSCLC
MET proto-oncogene receptor tyrosine kinase (MET), including via increased hepatocyte growth factor (HGF) ligand	Alectinib [134,137,138,139,140] Ceritinib and lorlatinib [79,98,117]	NSCLC
SRC proto-oncogene, non-receptor tyrosine kinase (SRC)	Crizotinib [141] Alectinib [138]	NSCLC
Discoidin domain receptor tyrosine kinase 2 (DDR2)	Alectinib [79]	NSCLC
Fibroblast growth factor receptor 2 (FGFR2)	Ceritinib [79]	NSCLC
ERb-B4 receptor tyrosine kinase 4 (ErbB4)	Lorlatinib [124]	Neuroblastoma
Interleukin 10 receptor subunit alpha (IL10RA)	Crizotinib [142] Alectinib [142] Brigatinib [142] Lorlatinib [142]	ALCL
Protein tyrosine phosphatase non-receptor tyrosine kinase 1/2 (PTPN1/2) loss	Crizotinib [143]	ALCL

**Table 3 cancers-13-06003-t003:** A skew towards immunosuppression in the tumour microenvironment can drive resistance to immune checkpoint inhibitors.

Event	Impact on the Tumour Microenvironment
Increased AXL receptor tyrosine kinase (AXL) expression	Increases regulatory T cells, MDSCs and M2 macrophages [185]
Increased Wnt signalling	Decreases tumour infiltrating lymphocytes [185]
Loss of Phosphatase and tensin homolog (PTEN)	Induces vascular endothelial growth factor (VEGF) production and reduces T cell infiltration [176,185]
Loss of functional beta 2 microglobulin	Dysfunctional CD8^+^ T cells [175,188]
Hypoxia	Dysfunctional CD8^+^ T cells [184,187]
Upregulation of T cell immunoglobulin and mucin-domain containing-3 (Tim-3)	Dysfunctional T helper 1 (Th1) cells and reduced cytokine expression [189,190]
Reduced expression of absent in melanoma 2 (AIM2)	Decreases inflammation [181]
Reduced expression of poliovirus receptor-related immunoglobulin domain containing protein (PVRIG)	Dysfunctional CD8^+^ T cells [181]
Increased expression of mannosidase alpha class 2A member 1 (MAN2A1)	Altered Th1/T-helper 2 (Th2) axis towards Th2 expression [181]

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
