# Peer review of "Resistance to Targeted Agents Used to Treat Paediatric ALK-Positive ALCL"

_cancers, 2021, doi:10.3390/cancers13236003_

Round 1

Reviewer 1 Report

Hare et al. provide an informative and updated review on the resistance mechanisms to targeted agents used for the therapy of pediatric ALK-positive anaplastic large cell lymphoma (ALCL). The review is weel organized and provides useful concepts for translational and clinical medicine of ALCL. A few suggestions for improvement are detailed below.

MAJOR ISSUES

  1. A figure depicting the structure of the ALK protein (with details of the functional domains) and the sites targeted by the different ALK inhibitors would be very helpful for a better understanding of resistance mutations.
  2. The authors state that ALK gene amplification has been reported as a mechanism of resistance to ALK inhibitors in NSCLC cell lines and patients and in ALCL cell lines (lines 270-276). What evidence is available in primary ALCL samples for ALK gene amplification as a mechanism of clinical resistance to ALK inhibitors?
  3. TP53 disruption is obviously a major mechanism of resistance to genotoxic agents, but, since it causes genetic instability, it may also favor accumulation of multiple mutations that, in some cases, finally confer resistance also to targeted agents. This has been well established in other lymphoid malignancies. The authors should mention these important concepts, comparing what is known in ALCL versus other lymphoid malignancies in which TP53-mediated refractoriness is understood in depth. In this respect, comparing the role of TP53 disruption in ALCL versus the case of chronic lymphocytic leukemia may be very informative (recent and useful references in this respect are Moia R, et al. Precision Medicine Management of Chronic Lymphocytic Leukemia. Cancers (Basel). 2020;12(3):642; Moia R, et al. Targeting p53 in chronic lymphocytic leukemia. Expert Opin Ther Targets. 2020;24(12):1239-1250

MINOR ISSUES

  1. For clarity sake, fig. 4 might be improved by representing both the cell cytoplasm and nucleus

Reviewer 2 Report

This manuscript reports on the rare entity of pediatric ALK+ anaplastic large cell lymphoma (ALCL) and attempts at summarizing potential therapeutic approaches. This is merely a review of published data indicating that there is no clear guideline of which type of treatment could be considered for these rare cases. The manuscript is providing a good overview of current shortcomings in this issue but does not provide any solution. The objective of this endeavor should be stated more clearly.

Round 2

Reviewer 1 Report

The authors have properly addressed the issues that had been raised.

Author Response

Many thanks for your time